# Hygiene Knowledge, Attitudes, and Behaviors of Greek Undergraduate Students on Mobile Phones and Corresponding Devices' Microbial Assessment

Dimitrios Delitzakis [1,*] , Vilelmine Carayanni [2], Kyriakoula Merakou [1] and Panagiota Giakkoupi [1]

1 Department of Public Health Policy, University of West Attica, 11521 Athens, Greece
2 Department of Tourism Management, University of West Attica, 12243 Athens, Greece
* Correspondence: ddelitzakis@uniwa.gr

**Abstract:** The purpose of this study was to investigate the knowledge, attitudes, and behaviors of Greek undergraduate students on hand hygiene and mobile phone hygiene, in relation to their mobile phones' microbial load. An anonymous self-reported questionnaire was distributed among 100 students and swabs were taken from their devices for microbiological cultures and for an on-site bioluminescence microbial load assessment. Hand washing was considered the most effective personal protective behavior by the majority (89%), while spatial restrictions (40%) and forgetfulness (31%) were the main reasons for neglecting hand washing. Most students washed their hands 6–10 times/day (32%) for approximately 11–20 s (35%). Although most devices were cleaned/disinfected within the last week (28%), many were contaminated with *Staphylococcus saprophyticus* (84%), *Staphylococcus aureus* (13%), and *Escherichia coli* (39%), while 75% of the samples exceeded the "fail" threshold limit of the luminometer's measurements. Moreover, statistically significant differences ($p < 0.05$) were found between the devices' microbial load and certain hand hygiene behaviors and preferences. The median cfu/cm$^2$ of *Escherichia coli* was higher among first-year students of health sciences, while *Staphylococcus aureus* was more prevalent in females' than in males' mobile phones. It w therefore understood that undergraduate students' knowledge, attitudes, and behaviors on hand hygiene and mobile phone hygiene are highly intertwined with the microbial load on their mobile phones.

**Keywords:** hand hygiene; mobile phones; undergraduate students; microbial load; ATP





## 1. Introduction

In 1938, Price found that the microorganisms detected on the hands can be divided into two distinct categories, namely resident and transient microbial flora. The resident flora is mainly found under the superficial cells of the stratum corneum, as well as on the surface of the hands. On the contrary, transient flora colonizes the superficial layers of the skin and it is easier to remove with normal hand hygiene [1]. Bacteria are the most dominant group of the human skin microbiome, including species of coagulase-negative staphylococci (CoNS), with *Staphylococcus epidermidis* being the dominant resident species, anaerobic *Propionibacterium acnes*, *Streptococcus*, *Micrococcus*, *Corynebacterium*, and *Acinetobacter*. Apart from the above bacterial species, the human skin is also colonized by various fungal species, such as *Candida*, *Malassezia*, *Cryptococcus*, etc. [2].

Particularly, the hand microbiome appears to be more dynamic over time, demonstrating a greater bacterial diversity (>3 times more bacterial phylotypes per individual), than other skin sites. Hence, bacterial communities on hands are generally enriched with different species over time, compared to other human body areas. Additionally, the palms of women demonstrate greater bacterial diversity than those of men, with *Propionibacteria* and *Corynebacterium* being more abundant on male hands, while *Enterobacteriales*, *Moraxellaceae*, *Lactobacillaceae*, and *Pseudomonaceae* are more abundant on female hands. However, the

diversity of fungal species on human hands is considered to be intermediate, compared to other skin sites of the human body [3]. Moreover, the distribution and density of the hands' microbiota depends on age and other environmental factors, such as the sebum secretion, pore blockage, as well as skin temperature and moisture. The existence of a healthy skin microbiota is based on the creation of a protective ecosystem against pathogens [4].

Human hands are the main means of interpersonal, intrapersonal, and environmental transfer of microorganisms through touch [3]. The transmissibility of transient flora depends on the type and number of microorganisms, as well as on the moisture of the skin [1]. Therefore, the hands of some individuals may be more prone to being colonized with microorganisms, such as *Staphylococcus aureus* and Gram negative bacteria such as *Escherichia coli*, or yeasts [1].

For instance, the hands of healthcare workers can be colonized with Gram positive microorganisms, such as *Staphylococcus aureus*. These microorganisms are the main cause of hospital-acquired infections (HAIs), including bacteremia, pneumonia, and other skin and soft tissue infections [5]. *Staphylococcus aureus* is considered to be a potentially pathogenic species, but it may be part of the normal skin and nasal mucosa flora in approximately 10–20% of healthy people [2]. Although the normal skin flora is less likely to be associated with infections, there is the potential for it to be responsible for infections in sterile body cavities, such as the eyes, or on intact skin [6]. Interestingly, the role that healthcare workers' hands play in the spread of HAIs has been thoroughly investigated, but there are a limited number of studies investigating the hand hygiene of undergraduate students.

On the other hand, inanimate surfaces or so called "fomites" can act as reservoirs of pathogens, contributing to the spread of various infectious diseases [7]. Mobile phones are a typical example of such fomite surfaces, since they are considered an essential "high-touch" handheld item in today's society [8]. They are part of the so-called "emotional technology" due to the personalized services provided through them, and are considered an integral accessory for both professional and private life [6]. As this advanced technology becomes widespread, many people do not consciously realize how often they touch their phones or where they place them [8].

Mobile phones can therefore act as a reservoir of microorganisms, especially when no cleaning or disinfection procedures are performed, increasing the risk of cross-contamination [9] between the user's skin and other surfaces or food [10]. The generated heat during their operation, or when placed inside a pocket, creates the ideal conditions for the incubation of pathogenic microorganisms [8]. Indicatively, several bacterial species have been identified on mobile phone surfaces, in healthcare, as well as in community settings. The most commonly isolated bacteria from mobile phone surfaces, which pose a potential health risk, are *Staphylococcus aureus*, CoNS (with *Staphylococcus epidermidis* and *Staphylococcus saprophyticus* being the most dominant species), and *Escherichia coli* [7].

It is also worth mentioning that even the complacency that the use of protective cases, films, and screen protectors on mobile phones reduces the microbial load of mobile phones, seems to have been disproven. The application of such accessories does not affect the levels of microbial colonization on mobile phones, since their use remains unchanged. Only the use of alcohol-based disinfectants can substantially reduce the number of colonies on mobile phone surfaces, by 75% [11].

As for the methods used, in order to assess the cleanliness of a surface, certain procedures such as visual inspections, microbial cultivations, as well as the application of bioluminescence tests are used [12]. These tests are based on measuring the light produced by a bioluminescence enzyme reaction, between luciferin–luciferase and adenosine triphosphate (ATP). ATP is an energy molecule found in all living cells, the presence of which indicates improper cleaning and the contamination of a surface. The amount of light produced from this reaction is proportional to the amount of a sample's ATP concentration. The light produced is measured by a luminometer, which displays the results in relative light units (RLUs) within a few seconds. Bioluminescence tests are widely used, due to their speed, ease of use, and cost-effectiveness [13]. However, the measurement of ATP

does not indicate the presence (or absence) of a particular microbial species, thus making it a quantitative indicator [8]. As a result, it should not be considered a substitute for classic routine microbiological culture methods [12].

Moreover, mobile phones are widely used by undergraduate students worldwide for social, as well as for academic purposes. Specifically, students who belong to faculties of health sciences tend to use their mobile phones during their clinical practice in hospitals, or in clinical laboratories. Similarly, students who attend curricula of other sciences also use their mobile phones during their classes or their internship in offices, where many people are usually present [14]. This constant handling of mobile phones by undergraduate students, in all places and occasions, exposes their device to an array of microorganisms [7]. Although the bacterial colonization of mobile phones in hospital settings has been thoroughly studied, there are only a few studies that investigated the contamination of mobile phones in the academic community [15]. Most of these studies have been conducted in underdeveloped and developing Asian and African countries. In the literature, there are a limited number of recent studies on mobile phone hygiene among undergraduate students, concerning the European countries, as summarized in Table 1.

**Table 1.** Recent European studies on mobile phone hygiene among undergraduate students.

| References | Objective of the Study | Key Highlights |
|---|---|---|
| Maurici et al., 2023 [16] | • Bacterial contamination of mobile phones among 83 healthcare university students in Italy, in relation to users' demographics, habits, and device characteristics.<br>• Evaluation of HPC 22 °C, HPC 37 °C, enterococci, Gram negative bacteria, and staphylococci. | • Bacterial loads of 416 cfu/dm$^2$ for HPC 37 °C and 442 cfu/dm$^2$ staphylococci, followed by HPC 22 °C, enterococci, and Gram negative bacteria.<br>• The majority of samples were positive for HPC 37 °C, HPC 22 °C, and staphylococci (98%), while enterococci (66%) and Gram negative bacteria (17%) were less frequent.<br>• Students with a daily internship attendance had higher HPC 22 °C bacterial loads than those attending <6 days/week. |
| Dubljanin et al., 2022 [17] | • Characterization of fungal contamination among the mobile phones of 492 medical students in Serbia.<br>• Investigation of mobile phones' usage and cleaning habits.<br>• Identification of independent risk factors for fungal contamination, and awareness of mobile phones as a potential route of infection. | • Fungal contamination was confirmed in 32.11% of mobile phone samples.<br>• *Candida albicans* was the most frequent fungal isolates on students' mobile phones (28.5%), followed by *Aspergillus niger* (11.4%) and *Penicillium chrysogenum* (9.5%).<br>• Lack of mobile phone cleaning (OR = 0.381; $p$ <0.001) and the use of mobile phones near patients' beds (OR = 0.571; $p$ = 0.007) were the independent risk factors associated with fungal contamination of students' mobile phones.<br>• Students who use their mobile phones in hospital wards had higher rates of fungal contamination. |

**Table 1.** *Cont.*

| References | Objective of the Study | Key Highlights |
|---|---|---|
| Sadiq et al., 2021 [18] | • Bacterial contamination prevalence on the mobile phones of 233 dental students in Cyprus.<br>• Identification of the bacterial isolates and assessment of their antimicrobial susceptibility patterns.<br>• Evaluation of the efficiency of the applied disinfectant. | • Microbial contamination was found in 81% (120.953 cfu/mL) of the swab samples taken, without prior use of alcohol-based wipes.<br>• Microbial contamination in swab samples taken after one-time disinfection reduced to 21% (201 cfu/mL).<br>• The most common microorganisms isolated were CoNS (69%) and *Aspergillus niger* (13%).<br>• All of the isolated bacteria were susceptible to all antibiotics used. |
| Cicciarella Modica et al., 2020 [19] | • Assessment of the microbiological contamination of mobile phones, among 108 students of healthcare sciences in Italy, in relation to their demographic characteristics and their mobile phones' handling habits. | • Staphylococci were present in 85% of mobile phones, followed by enterococci (37%) and coliforms (6.5%).<br>• *Escherichia coli* was never detected.<br>• *Staphylococcus epidermidis* was the most frequently isolated staphylococcal species (72%), followed by *S. capitis* (14%), *S. saprophyticus*, *S. warneri*, *S. xylosus* (6%), and *S. aureus* (4%).<br>• HPC 37 °C, ranged from $0–1.2×10^4$ cfu/dm$^2$ (mean = 362 cfu/dm$^2$).<br>• The male sex was only significantly associated with higher HPCs and enterococcal contamination. |

Considering all the above issues, further investigation is imperative among the social group of undergraduate students. Therefore, the present study aims to fill the gap of the understudied field of hand hygiene and mobile phone hygiene among undergraduate students. Unlike the previous studies, the novelty of this research lies in the combined investigation of the knowledge, attitudes, and behaviors of Greek undergraduate students, concerning their hand hygiene, as well as their mobile phone hygiene habits, in relation to the microbial load on their mobile phone devices, which was also assessed. It is worth mentioning that no microorganisms were directly isolated from students' hands, since that was not within the aims of this study. Instead, this study detected the most prevalent bacterial species on mobile phone surfaces, which are considered transient microbial flora on students' hands. This indirect approach was selected, in order to demonstrate the degree of interaction between hands and mobile phones, with an emphasis on removing the hands' transient flora through proper hand hygiene, as well as from the surfaces of students' mobile phones through adequate cleaning/disinfection procedures. The results of this study are considered of great importance, since they provide a clear viewpoint of the students' hygienic status and mindset, based on their sex, faculty, and year of study, where differences are expected, through the aspect of a European country like Greece. These findings are vital, especially in terms of public health policy making from the perspective of the current post-COVID circumstances.

## 2. Materials and Methods

### 2.1. Schematic Overview of the Study

This study consisted of two main sampling procedures, including the distribution of an anonymous self-reported questionnaire and a double microbiological sampling process from the participants' mobile phones, as presented in Figure 1.

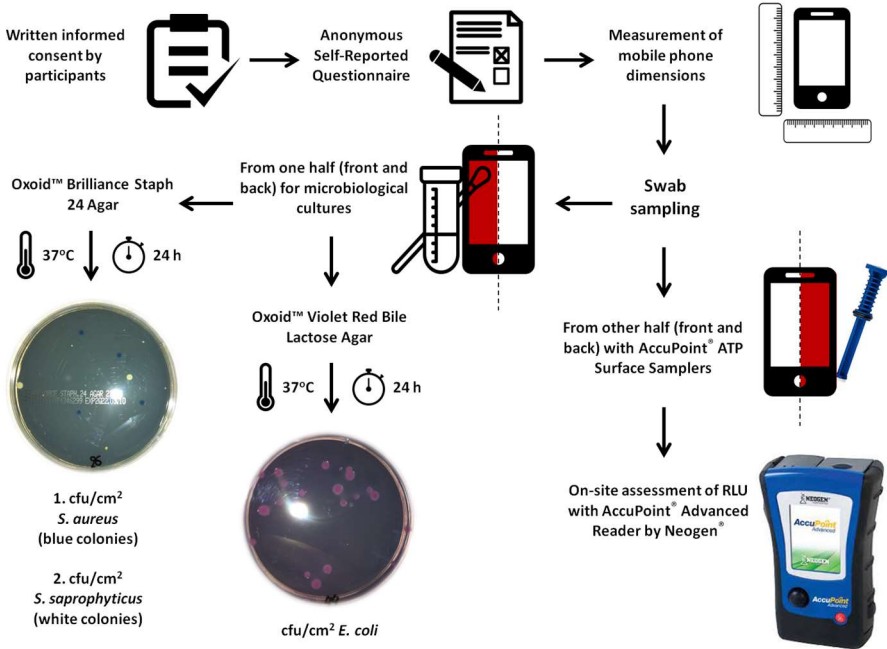

**Figure 1.** Schematic flow diagram concerning the procedures of the study.

The above flow diagram offers a visualization of the performed procedures and a better understanding of this study's objectives, regarding the investigation of the undergraduate students' knowledge, attitudes, and behaviors on hand hygiene and mobile phone hygiene, as well as the assessment of the microbial load on their mobile phone devices. Finally, it is worth mentioning that ethical approvals were required in order to handle the obtained microbiological samples; therefore, all procedures were conducted in accordance with the international standards, as described in the "Ethics Approval" and the "Institutional Review Board Statement" sections of this article.

### 2.2. Study Design and Setting

This cross-sectional study was conducted in the period between February and May 2022. The Alexandreia campus of the International Hellenic University, located in the region of Sindos, Thessaloniki, Greece, was selected as the setting to conduct the research. This campus consists of faculties of healthcare as well as non-healthcare sciences, which makes it the ideal setting for the purposes of this study.

### 2.3. Study Population

The study's sample consisted of 100 male and female undergraduate students, from healthcare as well as non-healthcare sciences, and from all academic years. The participation was voluntary, while the inclusion criteria were the following: undergraduate students whose age was at least 18 years of age; undergraduate students who studied at the Alexandreia campus of the International Hellenic University; undergraduate students who carried a mobile phone device at the time of participation; undergraduate students who did not clean or disinfect their mobile phones by any means right before their participation in the study; undergraduate students who consented to participate in the study. Therefore, individuals who did not meet all of the above criteria were excluded from participation in

this study. No further criteria were applied, taking into account parameters such as sex, faculty and years of studies. Considering the above criteria and parameters, the size of the sample was considered satisfactory for the purposes of this study [20].

The study was conducted on random and unannounced days and working hours of each week. Similarly, the study's sampling point was placed each time at different and randomly selected locations of the campus, while the convenience sampling method was performed. An information banner related to the objectives of the study was used, placed each time next to the study's sampling point, in order to attract passing students who were willing to participate. The purpose of the above tactics was to collect a representative sample of the student population and to minimize the "Hawthorne effect" [21]. Students might have cleaned their mobile phones right before their participation in the study if they knew in advance the time, date, and location of the sampling, leading to non-objective microbiological results.

### 2.4. Questionnaire

An anonymous questionnaire consisting of 40 questions was distributed to the participants, assessing their knowledge, attitudes, and behaviors on hand hygiene and mobile phone hygiene. The administered questionnaire was completed by self-report, the questions of which were separated into three distinct parts, as follows: Q1–Q5 for demographics (Part A); Q6–Q32 for hand hygiene (Part B); Q33–Q40 for mobile phone hygiene (Part C). The study's questionnaire was partially composed by a portion of the questions used in the research questionnaires of Głabska et al. and Guzek et al. on hand hygiene [22,23], as well as that of Cicciarella Modica et al. on mobile phone hygiene [19], after permission was obtained by the research teams' correspondence.

The questionnaire was initially translated from English to Greek, then from Greek to English, and finally from English to Greek, which is the native tongue of the study's participants. This triple translation method was applied in order to avoid any discrepancies and misunderstandings [24]. The questionnaire was then pilot tested on 10 participants who met the inclusion criteria to ensure its accuracy and understandability [25]. The estimated time required to complete the questionnaire was found to be approximately 10 min. At the time of the study, all questionnaires received a unique ascending number upon their completion by the participants.

### 2.5. Microbiological Procedures

After students filled in their questionnaires, two samples were taken from the mobile phone device of each participant: one sample intended for standard microbiological cultures, and another sample for an on-site assessment of the mobile phone's microbial load. Beforehand, the dimensions (height and width in centimeters) of each participant's mobile phone were measured and registered in the corresponding questionnaire. Samples were taken from the front surface and back surface of each mobile phone [26] by mentally dividing its surface into two equal parts along its vertical axis. The microbial load was considered to be evenly distributed on the surface of a mobile phone, thus the left or right side of each device was randomly chosen for each of the two samplings. If mobile phones had protective cases, films, or covers applied to them, then samples were taken from the screen and the outer surfaces of these accessories, since they were used in that state by their owners [27].

#### 2.5.1. Microbiological Sampling and Analysis

The samples for microbiological analysis were taken from the divided one half of the mobile phone surface, using sterile cotton swabs premoistened with saline water [26], by performing overlapping vertical and horizontal "S"-shaped rotating movements [28]. Swabs were then dipped in sterile vials containing 1 mL of saline water. Each vial received the same unique ascending number with the corresponding questionnaire of each mobile

phone's owner. Samples were kept in an isothermal cooling bag and were transferred within 2 h to a microbiological laboratory for analysis [29].

There, each sample vial was vortexed by a vortex mixer for approximately 60 s, and was inoculated into two selective culture mediums for the identification and enumeration of specific bacterial species of interest, which tend to be more abundant on mobile phones and constitute a potential health hazard. In particular, 100 µL from each sample were inoculated in sterile conditions onto Oxoid™ Brilliance Staph 24 Agar, which is a selective, chromogenic medium, validated in accordance with ISO 16140-6:2019 [30], for the isolation and enumeration of coagulase-positive *Staphylococcus aureus* and coagulase-negative *Staphylococcus saprophyticus*. In this selective media, *Staphylococcus aureus* colonies obtain a blue color, while the colonies of *Staphylococcus saprophyticus* appear white in color [31]. In addition, another 100 µL from each sample were inoculated in sterile conditions onto Oxoid™ Violet Red Bile Lactose Agar, which is a lactose-containing selective medium, validated in accordance with ISO 11133:2014/Amendment 2:2020 [32], for the detection and enumeration of *Escherichia coli*. In this selective medium, the colonies of *Escherichia coli* obtain a purple-pink color with purple halos [33].

Each pair of culture plates received the same unique ascending number with the corresponding questionnaire. All culture plates were incubated aerobically at 37 °C for 24 h. The enumeration of colonies, combined with the dimensions of the corresponding mobile phone (height in cm multiplied by width in cm), resulted in the calculation of cfu/cm$^2$ for each sample. Since two half sides (half front and half back) were sampled, the sum of these sides' dimensions matched the measured dimensions of each mobile phone. As a result, the calculated cfu/cm$^2$ was correctly expressed to the corresponding microbial load distribution of each device.

2.5.2. On-Site Microbial Load Assessment

The on-site quantitative assessment of the microbial load on the participants' mobile phone devices was evaluated with a bioluminescence testing procedure (refer to introduction), using the AccuPoint® Advanced Reader by Neogen®, Lansing, MI, USA. This luminometer is a lightweight, portable diagnostic tool designed to determine the level of a surface's hygiene [34]. The sampling was conducted with the help of the AccuPoint® Advanced ATP Surface Samplers by Neogen®, Lansing, MI, USA. These samplers consist of two main parts: the premoistened cotton swab for surface sampling and the main body (cartridge) of the sampler, containing the luciferin–luciferase enzymes.

Firstly, samplers were exposed to room temperature for at least one hour before sampling, according to the manufacturer's guidelines. Secondly, samples were taken from the remaining other half of the mobile phone surface by performing overlapping vertical and horizontal "S"-shaped movements [28] in an approximate area of 10 cm by 10 cm of each mobile phone [12]. Cotton swabs were then fully depressed vertically into the cartridges and mixed for two seconds in order to activate the bioluminescence enzyme reaction between the ATP molecules contained in each of the mobile phone samples and the luciferin–luciferase enzymes contained in each cartridge. Afterwards, the whole cartridge was inserted into the luminometer, where its sensor measured the amount of light produced, expressing the results in RLUs within a few seconds. Each result was finally registered in the participant's corresponding questionnaire.

As for the RLU value range limits, most manufacturing companies of bioluminescence tests provide two-step limits concerning the results, namely as "pass", "marginal", and "fail". Pass results are obtained when the ATP concentration is below the lower limit, indicating a clean surface. Marginal results are obtained when the ATP concentration is above the lower limit and below the upper limit, indicating that caution must be taken in adherence to cleaning procedures. Fail results are obtained when the ATP concentration is above the upper limit, indicating the ineffective cleaning of a surface [13]. In this study, a value of 149 RLUs was set as the highest limit for "pass" results, and a value of 300 RLUs

as the threshold for "fail" results, according to the instructions by Neogen®, Lansing, MI, USA [12]. Therefore, results between 150–299 RLUs were considered "marginal".

Through this sampling procedure, students were able to know the cleanliness degree of their mobile phones, expressed in numbers, at the time of their participation. Furthermore, the authors were able to obtain a primary view of the sample's hygiene status, as well as this additional layer of microbiological data for statistical analysis. It is worth mentioning that this tactic also gave students the motive to participate in this study, since they would immediately know their mobile phones' microbial load results, rather than waiting for the culture results, which they would not have been able to know or be informed about due to the anonymity of their participation.

### 2.6. Statistical Analysis

Statistical analysis was performed using IBM® Statistical Package for the Social Sciences (SPSS®) for Windows®, version 29. Descriptive analysis was performed using frequency and contingency tables, whereas non-normal quantitative variables of RLU and cfu/cm$^2$ were summarized by median and interquartile range (IQR) [21]. The Pearson's chi-squared test was used in order to evaluate the differences between qualitative variables based on sex, faculty, and year of study. Moreover, the microbial load relationship between RLUs and cfu/cm$^2$ was assessed by the Spearman's rho test [35]. In addition, the Mann–Whitney test was used in order to estimate the differences between non-normal quantitative variables of RLUs and cfu/cm$^2$ by sex and faculty. The non-parametric one-way analysis of variance test (Kruskal–Wallis test) was also used in order to test the homogeneity between the groups of the non-normal quantitative variables of RLUs and cfu/cm$^2$ by year of study [36]. The statistical significance level was set at $\alpha = 0.05$ (two-tailed).

### 2.7. Ethics Approval

This study's research protocol received approval from the Research Ethics Committee of the University of West Attica (approval protocol code: 99754/9 November 2021). Moreover, permission was granted by the Rectorate of the International Hellenic University of Greece in order to conduct the research on the premises of the Alexandreia campus, located in the Sindos region of Thessaloniki, Greece. Permission was also granted by the administration of the microbiological laboratory, in which the microbiological analysis of the samples was conducted. Furthermore, participation in this study was voluntary and participants were informed about the procedures. All procedures were written in detail on the first pages of the informed consent document, provided along with the questionnaire pack. Written informed consent was received from each participant. In addition, the self-reported questionnaires distributed were anonymous, and neither any personally identifiable information nor the participants' medical history was asked. Finally, the anonymity of the participants was ensured in every stage of the study, and all collected data were confidentially used only for the purposes of the study research.

## 3. Results

### 3.1. Descriptive Statistics

#### 3.1.1. Demographic Description

One hundred undergraduate students (51 males and 49 females) of healthcare (55%) as well as non-healthcare (45%) sciences participated in this study. The majority of participants were in their final (fourth or greater) year of study (35%), followed by first-year (24%), second-year (22%), and third-year (19%) students. Additionally, most students (69%) used public transportation, while the remaining used private vehicles (30%) and walking (1%), in order to reach the campus. Moreover, most participants reached the university campus on all five working days per week (34%), followed by those who were present four times/week (26%), three times/week (26%), twice/week (9%), once/week (4%), or having another attendance frequency (1%).

### 3.1.2. Hand Hygiene

Concerning the questions in which the students' knowledge, attitudes, and behaviors on hand hygiene were assessed, the results are summarized in Table 2. Most of the participants considered the use of face masks and staying at home as equally effective (37%) protective measures that promote hand hygiene. The majority of students also preferred hand washing (72%), soap use (41%), liquid soap (47%), and paper towels (51%) as more effective methods and means of hand hygiene.

**Table 2.** The declared knowledge and preferences of the studied sample of undergraduate students on hand hygiene.

| Variable | | *n* (%) |
|---|---|---|
| Staying at home vs. wearing face mask | Staying at home | 18 (18%) |
| | Wearing face mask | 28 (28%) |
| | They are equally effective | 37 (37%) |
| | I do not know which one is better | 17 (17%) |
| Hand washing vs. wearing gloves | Hand washing | 72 (72%) |
| | Wearing gloves | 5 (5%) |
| | They are equally effective | 21 (21%) |
| | I do not know which one is better | 2 (2%) |
| Using soap vs. using alcohol disinfectant | Using soap | 41 (41%) |
| | Using alcohol disinfectant | 14 (14%) |
| | They are equally effective | 39 (39%) |
| | I do not know which one is better | 6 (6%) |
| Using liquid soap vs. using soap bar | Using liquid soap | 47 (47%) |
| | Using soap bar | 6 (6%) |
| | They are equally effective | 29 (29%) |
| | I do not know which one is better | 18 (18%) |
| Using paper towel vs. using hand dryer | Using paper towel | 51 (51%) |
| | Using hand dryer | 19 (19%) |
| | They are equally effective | 21 (21%) |
| | I do not know which one is better | 9 (9%) |
| How long do you think it takes to wash your hands? | Less than 5 s | 2 (2%) |
| | 5–10 s | 14 (14%) |
| | 11–20 s | 35 (35%) |
| | 21–40 s | 29 (29%) |
| | More than 40 s | 20 (20%) |
| | It does not matter | 0 (0%) |
| | I do not know | 0 (0%) |

In terms of personal protective behaviors, the students' responses are shown in Table 3. Hand washing was considered by 89% of the undergraduate students as the most effective personal protective behavior, followed by the use of face masks (83%), and the use of alcohol disinfectants (76%). As for the students' daily hand washing frequency, the results are presented in Figure 2. The majority of students (32%) washed their hands 6–10 times/day, followed by those who did so 3–5 times/day (23%), and 16–20 times/day (19%).

The declared reasons for neglecting hand washing are presented in Table 4. While most students stated that they always washed their hands (41%), hand washing was sometimes skipped, due to restrictions (40%), or because they neglected washing their hands (31%).

**Table 3.** The declared personal protective behaviors of the studied sample of undergraduate students.

| Variable | | *n* | % | % of Cases |
|---|---|---|---|---|
| | Staying at home | 28 | 6.4% | 28% |
| | Using face mask | 83 | 19% | 83% |
| | Avoid touching my face | 61 | 14% | 61% |
| | Wearing gloves | 5 | 1.1% | 5% |
| | Hand washing | 89 | 20.4% | 89% |
| Personal protective behaviors | Using alcohol disinfectant | 76 | 17.4% | 76% |
| | Avoid touching sick people | 68 | 15.6% | 68% |
| | Avoiding public places | 15 | 3.4% | 15% |
| | Medications or dietary supplements | 11 | 2.5% | 11% |
| | Other * | 0 | 0% | 0% |
| | Total | 436 | 100% | |

* If this answer was selected, then a blank space was provided, and students were asked to define.

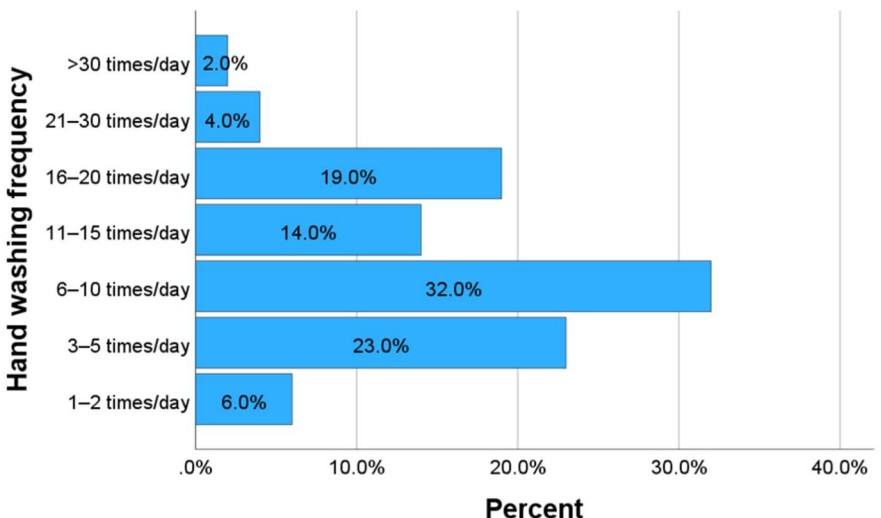

**Figure 2.** The declared hand washing frequency of the studied sample of undergraduate students.

**Table 4.** The declared reasons for neglecting hand washing by the studied sample of undergraduate students.

| Variable | | *n* | % | % of Cases |
|---|---|---|---|---|
| | There is no need to do it | 1 | 0.8% | 1% |
| | I do not feel like doing it | 3 | 2.4% | 3% |
| | Due to restrictions (e.g., no soap, no bathroom nearby) | 40 | 32.3% | 40% |
| Reasons for neglecting hand washing | Due to side effects (e.g., skin issues) | 6 | 4.8% | 6% |
| | I have no time to do it | 1 | 0.8% | 1% |
| | I forget to | 31 | 25% | 31% |
| | I always wash my hands | 41 | 33.1% | 41% |
| | Other * | 1 | 0.8% | 1% |
| | Total | 124 | 100% | |

* If this answer was selected, then a blank space was provided, and students were asked to define.

### 3.1.3. Mobile Phone Hygiene

Regarding the questions in which the students' knowledge, attitudes, and behaviors on mobile phone hygiene were assessed, the results are summarized in Table 5. All participants (100%) owned touchscreen smartphones, which were used by most of them (92%) during classes and other academic activities. However, the majority of undergraduate students (94%) avoided touching their mobile phones when they wore gloves. In addition, most

devices (67%) were equipped with a back cover, and sometimes (62%) participants used other accessories as well, such as headphones, Bluetooth devices, etc. Finally, 28% of mobile phone devices were last cleaned/disinfected by their owners about one week before the time of the study.

**Table 5.** The declared responses by the studied sample of undergraduate students, on questions concerning mobile phone hygiene.

| Variable | | *n* (%) |
|---|---|---|
| Choose your mobile phone type | With touchscreen | 100 (100%) |
| | With keypad | 0 (0%) |
| Do you use your mobile phone during classes? | Yes | 92 (92%) |
| | No | 8 (8%) |
| When was the last time you cleaned/disinfected your cell phone (approximately)? | Today | 12 (12%) |
| | One week ago | 28 (28%) |
| | One month ago | 27 (27%) |
| | One year ago | 5 (5%) |
| | Other * | 28 (28%) |
| How do you carry your mobile phone? | Without a case/cover | 14 (14%) |
| | In a protective flip case | 19 (19%) |
| | In a protective back cover | 67 (67%) |
| Do you use accessories (e.g., headphones, handsfree, Bluetooth, etc.), while using your mobile phone? | Never | 10 (10%) |
| | Sometimes | 62 (62%) |
| | Always | 28 (28%) |
| Do you touch your mobile phone, while wearing disposable gloves? | No | 94 (94%) |
| | Yes, without changing my gloves | 2 (2%) |
| | Yes, but I change my gloves right after | 4 (4%) |

\* If this answer was selected, then a blank space was provided, and students were asked to define.

### 3.2. Microbiological Assessment

Growth of bacterial colonies was observed in 85% of the total number of culture plates of Brilliance Staph 24 Agar (Figure 3a), with a median value of 0.27 (IQR: 0.63) cfu/cm$^2$. In particular, white-colored colonies of *Staphylococcus saprophyticus* were formed in 84% of the total number of culture plates, with a median value of 0.34 (IQR: 0.58) cfu/cm$^2$. However, blue-colored colonies were developed in 13% of the total number of culture plates, with a median value of 0.11 (IQR: 0.09) cfu/cm$^2$, indicating the presence of *Staphylococcus aureus*. The Kruskal–Wallis test indicated statistically significant differences in the total median cfu/cm$^2$ of the Brilliance Staph 24 Agar culture results between participants' means of transport categories ($p = 0.038$), as well as between certain personal protective behaviors, such as abstaining from touching their face ($p = 0.007$), evading sick people ($p = 0.014$), and hand washing after blowing their nose ($p < 0.001$), as shown in Table 6.

Similarly, significant differences were also found between the median cfu/cm$^2$ of *Staphylococcus saprophyticus*, grouped by the same aforementioned variables, concerning the participants' means of transport ($p = 0.033$), abstaining from touching their face ($p = 0.007$), evading sick people ($p = 0.022$), and hand washing after blowing their nose ($p < 0.001$), as presented in Table 7.

The same test revealed statistically significant differences between the median cfu/cm$^2$ of *Staphylococcus aureus* regarding participants' attendance frequency ($p = 0.004$), soap type preference ($p = 0.040$), and evading sick people ($p = 0.045$), as demonstrated in Table 8.

In addition, *Escherichia coli* was isolated in 39% of the total number of culture plates of Violet Red Bile Lactose Agar (Figure 3b), with a median value of 0.25 (IQR: 0.32) cfu/cm$^2$. The Kruskal–Wallis test indicated significant differences between the median cfu/cm$^2$ of *Escherichia coli*, the participants' soap type preference ($p = 0.006$), and hand washing among the preferred personal protective behaviors ($p = 0.016$), as presented in Table 9.

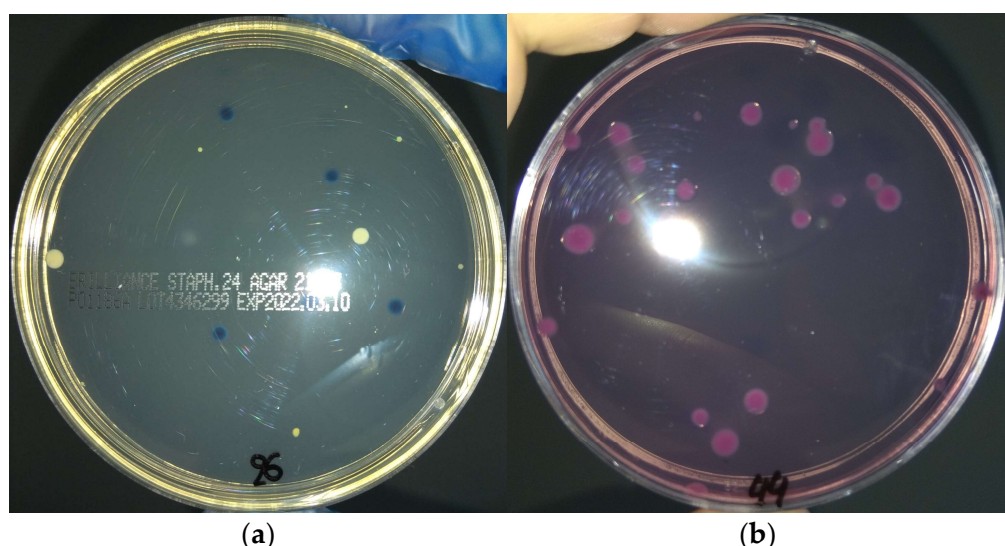

**Figure 3.** Microbiological culture plates of: (**a**) Brilliance Staph 24 Agar, where white-colored colonies of *Staphylococcus saprophyticus* and blue-colored colonies of *Staphylococcus aureus* were developed; (**b**) Violet Red Bile Lactose agar, where purple-pink-colored colonies of *Escherichia coli* were developed.

**Table 6.** Statistically significant variables, based on the total cfu/cm$^2$ of Brilliance Staph 24 Agar.

| Variable | cfu/cm$^2$ (Brilliance Staph 24 Agar) | *p* |
|---|---|---|
| What means of transport did you use today to come to the university? | | |
| Private vehicle | 0.53 (IQR: 0.69) | |
| Public transport | 0.22 (IQR: 0.50) | 0.038 ** |
| Other * | 2.41 (IQR: 0.00) | |
| Avoid touching my face (personal protective behaviors) | | |
| Yes | 0.23 (IQR: 0.38) | |
| No | 0.58 (IQR: 0.80) | 0.007 ** |
| Avoid touching sick people (personal protective behaviors) | | |
| Yes | 0.38 (IQR: 0.65) | |
| No | 0.18 (IQR: 0.24) | 0.014 ** |
| I wash my hands after blowing my nose | | |
| Never | 1.38 (IQR: 1.46) | |
| Sometimes | 0.30 (IQR: 0.59) | <0.001 ** |
| Always | 0.23 (IQR: 0.44) | |

* If this answer was selected, then a blank space was provided, and students were asked to define. ** *p*-values were calculated using the Kruskal–Wallis test.

Finally, the on-site bioluminescence measurement of ATP on participants' mobile phones resulted in a median value of 711.50 (IQR: 980.75) RLUs. Only 12% of the students' mobile phones presented a "pass" microbial load under 150 RLUs, while the majority of 75% exceeded the "fail" threshold of 300 RLUs, and 13% were considered "marginal". According to the Kruskal–Wallis test, differences were observed between the median RLU results and participants' hand washing frequency (*p* = 0.030), soap type preference (*p* = 0.037), and their hand washing reluctance (*p* = 0.017), as summarized in Table 10.

**Table 7.** Statistically significant variables, based on the cfu/cm$^2$ of *Staphylococcus saprophyticus*.

| Variable | cfu/cm$^2$ (*Staphylococcus saprophyticus*) | *p* |
|---|---|---|
| What means of transport did you use today to come to the university? | | |
| Private vehicle | 0.53 (IQR: 0.66) | |
| Public transport | 0.30 (IQR: 0.41) | 0.033 ** |
| Other * | 2.41 (IQR: 0.00) | |
| Avoid touching my face (personal protective behaviors) | | |
| Yes | 0.31 (IQR: 0.40) | |
| No | 0.52 (IQR: 0.87) | 0.007 ** |
| Avoid touching sick people (personal protective behaviors) | | |
| Yes | 0.43 (IQR: 0.69) | |
| No | 0.23 (IQR: 0.36) | 0.022 ** |
| I wash my hands after blowing my nose | | |
| Never | 1.38 (IQR: 1.46) | |
| Sometimes | 0.33 (IQR: 0.51) | <0.001 ** |
| Always | 0.24 (IQR: 0.37) | |

* If this answer was selected, then a blank space was provided, and students were asked to define. ** *p*-values were calculated using the Kruskal–Wallis test.

**Table 8.** Statistically significant variables, based on the cfu/cm$^2$ of *Staphylococcus aureus*.

| Variable | cfu/cm$^2$ (*Staphylococcus aureus*) | *p* |
|---|---|---|
| How often do you attend the University premises during the week? | | |
| Once a week | 0.00 (IQR: 0.00) | |
| Twice a week | 0.00 (IQR: 0.00) | |
| Three times a week | 0.08 (IQR: 0.00) | 0.004 ** |
| Four times a week | 0.12 (IQR: 0.00) | |
| Five times a week | 0.11 (IQR: 0.09) | |
| Other * | 0.00 (IQR: 0.00) | |
| Using liquid soap vs. using soap bar | | |
| Using liquid soap | 0.08 (IQR: 0.06) | |
| Using soap bar | 0.12 (IQR: 0.08) | |
| They are equally effective | 0.13 (IQR: 0.21) | 0.040 ** |
| I do not know which one is better | 0.24 (IQR: 0.00) | |
| Avoid touching sick people (personal protective behaviors) | | |
| Yes | 0.12 (IQR: 0.09) | |
| No | 0.08 (IQR: 0.00) | 0.045 ** |

* If this answer was selected, then a blank space was provided, and students were asked to define. ** *p*-values were calculated using the Kruskal–Wallis test.

**Table 9.** Statistically significant variables, based on the cfu/cm$^2$ of *Escherichia coli*.

| Variable | cfu/cm$^2$ (*Escherichia coli*) | *p* |
|---|---|---|
| Using liquid soap vs. using soap bar | | |
| Using liquid soap | 0.08 (IQR: 0.38) | |
| Using soap bar | 0.00 (IQR: 0.43) | 0.006 * |
| They are equally effective | 0.00 (IQR: 0.00) | |
| I do not know which one is better | 0.00 (IQR: 0.16) | |
| Hand washing (personal protective behaviors) | | |
| Yes | 0.00 (IQR: 0.11) | |
| No | 0.10 (IQR: 0.40) | 0.016 * |

* *p*-values were calculated using the Kruskal–Wallis test.

**Table 10.** Statistically significant variables, based on the RLU results.

| Variable | RLU | *p* |
|---|---|---|
| Hand washing frequency | | |
| 1–2 times/day | 1153 (IQR: 1741) | |
| 3–5 times/day | 1096 (IQR: 1589) | |
| 6–10 times/day | 660 (IQR: 1292.25) | |
| 11–15 times/day | 747 (IQR: 590.25) | 0.030 * |
| 16–20 times/day | 334 (IQR: 553.5) | |
| 21–30 times/day | 672 (IQR: 416.5) | |
| >30 times/day | 1180 (IQR: 641) | |
| Using liquid soap vs. using soap bar | | |
| Using liquid soap | 539 (IQR: 594) | |
| Using soap bar | 373 (IQR: 368) | 0.037 * |
| They are equally effective | 1208 (IQR: 1561) | |
| I do not know which one is better | 941 (IQR: 940.75) | |
| I do not feel like doing it (Reasons for neglecting hand washing) | | |
| Yes | 2890 (IQR: 1682) | 0.017 * |
| No | 707 (IQR: 915.5) | |

* *p*-values were calculated using the Kruskal–Wallis test.

However, no statistical association between RLUs and cfu/cm$^2$ was found, according to the results of Spearman's rho test (*p* >0.05). Indicatively, correlations between the RLU and cfu/cm$^2$ microbiological results, grouped by RLU limit range values and the isolated bacterial species, are demonstrated in Table 11.

**Table 11.** Correlations between the RLU and cfu/cm$^2$ microbiological results.

| RLU | *Staph. aureus* (cfu/cm$^2$) | *Staph. saprophyticus* (cfu/cm$^2$) | *E. coli* (cfu/cm$^2$) |
|---|---|---|---|
| ≤149 (Pass) | 0.10 (IQR: 0.08) | 0.24 (IQR: 0.00) | 0.10 (IQR: 0.09) |
| 150–299 (Marginal) | 0.32 (IQR: 0.60) | 0.25 (IQR: 0.36) | 0.37 (IQR: 0.70) |
| ≥300 (Fail) | 0.42 (IQR: 1.33) | 0.16 (IQR: 0.09) | 0.23 (IQR: 0.24) |

*3.3. Influence of Sex, Faculty, and Year of Study*

Faculty was found to be the greatest determinant in terms of the microbial load on students' mobile phones, as shown in Table 12. Statistically compelling variations were observed in the median RLU values (*p* < 0.001), as well as in the median cfu/cm$^2$ values (*p* < 0.001) of *Escherichia coli* between undergraduate students of health sciences and other sciences. Moreover, faculty influenced certain personal protective behaviors, such as the use of face masks or staying at home (*p* = 0.001), and soap type preference (*p* = 0.005).

**Table 12.** Statistically significant variables, based on the participants' faculty.

| Variable | Health Sciences | Other Sciences | *p* |
|---|---|---|---|
| Staying at home vs. wearing face mask * | | | |
| Staying at home | 11 (11%) | 7 (7%) | |
| Wearing face mask | 19 (19%) | 9 (9%) | 0.001 ** |
| They are equally effective | 23 (23%) | 14 (14%) | |
| I do not know which one is better | 2 (2%) | 15 (15%) | |
| Using liquid soap vs. using soap bar * | | | |
| Using liquid soap | 34 (34%) | 13 (13%) | |
| Using soap bar | 3 (3%) | 3 (3%) | 0.005 ** |
| They are equally effective | 10 (10%) | 19 (19%) | |
| I do not know which one is better | 8 (8%) | 10 (10%) | |
| RLU | 442 (IQR: 587) | 1008 (IQR: 1111) | <0.001 *** |
| cfu/cm$^2$ (*Escherichia coli*) | 0.08 (IQR: 0.34) | 0.00 (IQR: 0.04) | <0.001 *** |

* Some categories were merged due to low expected frequencies (<5). ** *p*-values were calculated using the Pearson's chi-squared test. *** *p*-values were calculated using the Mann–Whitney test.

Accordingly, significant differences were observed in the median cfu/cm$^2$ values ($p < 0.001$) of *Escherichia coli* based on participants' year of study. Students' year of study also influenced their soap type preference ($p = 0.020$), as well as the application of protective covers ($p = 0.002$) on their mobile phones, as summarized in Table 13.

**Table 13.** Statistically significant variables, based on the participants' year of study.

| Variable | 1st year | 2nd year | 3rd year | 4th + year | *p* |
|---|---|---|---|---|---|
| Using liquid soap vs. using soap bar * | | | | | |
| Using liquid soap | 14 (14%) | 8 (8%) | 14 (14%) | 11 (11%) | |
| Using soap bar | 4 (4%) | 1 (1%) | 0 (0%) | 1 (1%) | |
| They are equally effective | 3 (3%) | 8 (8%) | 3 (3%) | 15 (15%) | 0.020 ** |
| I do not know which one is better | 3 (3%) | 5 (5%) | 2 (2%) | 8 (8%) | |
| How do you carry your mobile phone? * | | | | | |
| Without a case/cover | 2 (2%) | 3 (3%) | 4 (4%) | 5 (5%) | |
| In a protective flip case | 2 (2%) | 3 (3%) | 1 (1%) | 13 (13%) | 0.002 ** |
| In a protective back cover | 20 (20%) | 16 (16%) | 14 (14%) | 17 (17%) | |
| cfu/cm$^2$ (*Escherichia coli*) | 0.16 (IQR: 0.40) | 0.00 (IQR: 0.09) | 0.09 (IQR: 0.34) | 0.00 (IQR: 0.00) | <0.001 *** |

* Some categories were merged due to low expected frequencies (<5). ** *p*-values were calculated using the Pearson's chi-squared test. *** *p*-value was calculated using the Kruskal–Wallis test.

As presented in Table 14, participants' sex mainly influenced the duration of hand washing ($p = 0.036$) and their preference between paper towel and hand dryer ($p = 0.020$). In addition, concerning the students' personal protective behaviors, abstaining from touching their face differed substantially ($p = 0.036$) between male and female participants, along with their hand washing forgetfulness ($p = 0.007$), as one of the stated reasons for neglecting hand washing. Finally, differences were also observed between the presence of blue colonies of *Staphylococcus aureus* in Brilliance Staph 24 Agar culture results, grouped by participants' sex ($p = 0.031$). However, the median cfu/cm$^2$ of *Staphylococcus aureus* was higher in the mobile phone samples of males (0.15 (IQR: 0.10) cfu/cm$^2$) than in the samples of female participants (0.10 (IQR: 0.11) cfu/cm$^2$), although this difference was not considered statistically significant ($p = 0.178$).

**Table 14.** Statistically significant variables, based on the participants' sex.

| Variable | Males | Females | *p* |
|---|---|---|---|
| Using paper towel vs. using hand dryer * | | | |
| Using paper towel | 27 (27%) | 24 (24%) | |
| Using hand dryer | 9 (9%) | 10 (10%) | |
| They are equally effective | 14 (14%) | 7 (7%) | 0.020 ** |
| I do not know which one is better | 1 (1%) | 8 (8%) | |
| How long do you think it takes to wash your hands? * | | | |
| Less than 5 s | 1 (1%) | 1 (1%) | |
| 5–10 s | 7 (7%) | 7 (7%) | |
| 11–20 s | 21 (21%) | 14 (14%) | |
| 21–40 s | 18 (18%) | 11 (11%) | 0.036 ** |
| More than 40 s | 4 (4%) | 16 (16%) | |
| It does not matter | 0 (0%) | 0 (0%) | |
| I do not know | 0 (0%) | 0 (0%) | |
| Avoid touching my face (Personal protective behaviors) | | | |
| Yes | 26 (26%) | 35 (35%) | |
| No | 25 (25%) | 14 (14%) | 0.036 ** |

**Table 14.** *Cont.*

| Variable | Males | Females | *p* |
|---|---|---|---|
| I am forgetting it (Reasons for neglecting hand washing) | | | |
| Yes | 22 (22%) | 9 (9%) | 0.007 ** |
| No | 29 (29%) | 40 (40%) | |
| Blue colonies of *Staphylococcus aureus* in Brilliance Staph 24 Agar culture results | | | |
| Presence | 3 (3%) | 10 (10%) | 0.031 ** |
| Absence | 48 (48%) | 39 (39%) | |

* Some categories were merged due to low expected frequencies (<5). ** *p*-values were calculated using the Pearson's chi-squared test.

## 4. Discussion

This study assessed the knowledge, attitudes, and behaviors of Greek undergraduate students on hand hygiene and mobile phone hygiene in a university setting, using an anonymous self-reported questionnaire. There was an equal share of male and female participants, as well as of students of health and other sciences, from all years of study. Most of the participants reached the university campus on all working days of each week via public transport. This situation is a cause for concern, since there is an increased health risk faced by students, due to overcrowding and frequent contact with various surfaces, as indicated by a previous study [37].

Taking into account the participants' hand hygiene assessment, most students preferred hand washing, soap use, paper towels, and liquid soap, rather than gloves, alcohol disinfectants, hand dryer, and soap bar (refer to Table 2). These findings are in agreement with the results of a previous study by Guzek et al., where the same questions were asked concerning the participants' knowledge and beliefs on proper hand hygiene and personal protection [23]. It seems by the declared answers that the participants of this study are well oriented, as far as the methods and means of hand hygiene. Speaking of hand hygiene, alcohol-based hand disinfectants, especially at concentrations between 60% and 80%, can significantly reduce the bacterial colonies on the skin, but they are incapable of eliminating most spores [38], while soap washes away bacteria by dissolving the oily layer on the skin's surface [39]. There are still ongoing debates, however, regarding the effectiveness of alcohol-based hand disinfectants and soap, which are used to ward off pathogens [38].

In addition, having studied the personal protective behaviors (refer to Table 3), hand washing had the highest share among respondents, as also proved in the study by Guzek et al. [23], followed by the use of face masks and alcohol disinfectants. Less popular personal protective behaviors were the avoidance of public places, medications or dietary supplements, and the use of disposable gloves. As for the participants' hand washing frequency (refer to Figure 2), although most students washed their hands frequently, there was still a considerable number of students who washed their hands inadequately. Previous studies demonstrated similar results, where the daily hand washing frequency of most students also ranged between 6–10 times [40,41]. Therefore, it is not enough just to prefer hand washing as a hand hygiene method, but it should also be performed with sufficient frequency as well.

However, the duration of hand washing by the majority of undergraduate students is considered adequate (refer to Table 2). Hands should be washed for at least 15 s, while making sure that the entire hands' surface is properly cleaned [42]. Additionally, while most participants always washed their hands no matter what, certain restrictions such as the absence of soap, or the proximity and accessibility of bathrooms, as well as the students' forgetfulness, were the main declared reasons why they neglected washing their hands (refer to Table 4). Other reasons (apart from the available choices) for neglecting hand washing, included participants' boredom. Similar trends were observed in previous studies [40], indicating forgetfulness [41,43,44], as well as the above restrictions [45], as the main reasons for neglecting hand washing. In order to solve this issue, students should

have easy access to hand hygiene means and facilities, as well as find efficient ways to remind them and keep them motivated to wash their hands.

There is a strong association of colonization between the hands and surfaces. As demonstrated by the findings of a study that analyzed a total of 69 pairs of samples obtained from hands and surfaces, in all cases, the same bacterial species was recovered, both from the hand and the respective environmental surface from which the sample was taken. Among the recovered bacteria, 81% were Gram negative rods, while 19% were Gram positive cocci [46]. This interconnectedness between the hands and surfaces is a cause for concern, especially when it comes to the spread of pathogens.

Concerning the students' mobile phone hygiene assessment (refer to Table 5), it is worth mentioning that all the participants in this study owned a touchscreen mobile phone device. Thus, the results of this study are not applicable to owners of keypad mobile phones. Moreover, protective back covers were applied to most participants' mobile phones, along with the use of other accessories, such as headphones, handsfree, Bluetooth, etc. While most students used their mobile phones during classes, almost no one touched their devices while wearing disposable gloves. Finally, the majority of mobile phones were last cleaned by their owners approximately one week to one month prior to the time of participation. The above findings coincide with those of a previous study by Cicciarella Modica et al., where the same questions were assessed [19]. However, other responses (apart from the available choices) indicated that a high share of mobile phones were never cleaned or last cleaned at their time of purchase. Similar response rates were also observed in the study by Ahmad et al., where 22.4% of participants cleaned their phones daily, 27% weekly, and 19.7% once a month [47]. Hence, it is safe to say that students need to clean/disinfect their mobile phones more frequently, particularly after attending classes or visiting crowded places.

The microbial load on participants' mobile phone devices was also assessed in this study, using conventional microbiological culture methods, as well as an on-site bioluminescence ATP measurement technique. In general, higher rates of *Staphylococcus* spp. were expected in the mobile phone samples, since it is part of the normal flora of the human skin [6]. Thus, the mobile phone devices of those who did not avoid touching their face as a personal protective behavior, as well as those who never washed their hands after blowing their nose, demonstrated a higher microbial load of total cfu/cm$^2$ in their Brilliance Staph 24 Agar culture results (refer to Table 6). Similar culture results were observed among the mobile phone samples of participants who reached the university campus on foot or with a private vehicle, as well as surprisingly, among those who avoided touching sick people. It is worth mentioning that this study's bacterial growth rate results match the results of the study by Cicciarella Modica et al., where 85% of the samples from 108 mobile phones of students of health sciences were also positive for staphylococci [19].

In particular, the aforementioned variables concerning the participants' means of transport, abstaining from touching their face, evading sick people, as well as hand washing after blowing their nose, demonstrated the same statistical significance tendency as above, in correlation with the cfu/cm$^2$ of *Staphylococcus saprophyticus* (refer to Table 7). Furthermore, a higher bacterial growth rate of *Staphylococcus aureus* was observed among the mobile phone samples of students who reached the university campus 4 or 5 times a week, as well as among those who were unsure about their soap type preference, and surprisingly, among those who avoided touching sick people (refer to Table 8). Accordingly, in a study sample of 27 mobile phones of secondary school students, the presence of potentially pathogenic microorganisms, including strains of *Staphylococcus aureus*, were also found [15]. It is therefore understood that certain everyday habits and behaviors, induce the colonization of students' mobile phones with opportunistic pathogenic bacteria.

In this study, *Staphylococcus* spp. (namely those of *Staphylococcus saprophyticus* and *Staphylococcus aureus*) were the most prevalent bacteria, followed by *Escherichia coli*, as also observed by over a third of the previous studies conducted both in healthcare and community settings [48–51]. For instance, in a study conducted in a Turkish hospital, higher levels of *Staphylococcus aureus* and *Escherichia coli* were observed on the hands of

60 intensive care unit nurses after ending their shift, compared to the beginning of their shift [52]. Another study also showed that the hands of nursing staff with skin disease are twice as likely to be colonized with Gram positive bacteria, such as *Staphylococcus aureus*, and Gram negative bacteria that are resistant to third-generation cephalosporins, such as *Escherichia coli* [53]. However, in a study among 66 students of health professions aged 19–22 years old, 9.1% of the participants' hands tested positive for *Escherichia coli* colonization before their clinical practice, while after their clinical practice, this percentage decreased to 6.1% of the participants [5].

The mobile phone devices of those who did not select hand washing as a personal protective behavior demonstrated a higher microbial load of *Escherichia coli* in $cfu/cm^2$. Similar results were observed among those who preferred the use of liquid soap, instead of soap bar, and those who did not know which of these was more effective (refer to Table 9). Since *Escherichia coli* is part of the normal intestinal flora, its presence on hands and consequently on mobile phones indicates fecal contamination due to poor personal and hand hygiene [54], especially after visiting the restroom. Accordingly, although *Staphylococcus saprophyticus* is part of the normal human flora of the perineum, rectum, urethra, cervix, and gastrointestinal tract, it is associated with lower urinary tract infections (UTIs) in young and middle-aged women [19]. Thus, its presence on human hands can only be considered transient flora, indicating poor personal and hand hygiene as well.

The bacterial survival on surfaces ranges from hours to months, depending on a variety of factors, such as the environmental conditions, the bacterial strain properties, as well as the characteristics of the surface itself. In the study by Simmonds–Cavanagh, the viability of a variety of pathogenic bacteria on mobile phones in healthcare was assessed. *Bacillus cereus*, followed by CoNS, *Acinetobacter baumannii*, *Enterococcus faecalis*, and *Staphylococcus aureus* were found to be the most successful pathogenic bacteria to remain viable on mobile phones at the 28-day cut-off. On the other hand, *Escherichia coli* and *Pseudomonas aeruginosa* remained viable on mobile phones and were last detected at 6 h, while *Pseudomonas strutzeri* was last detected at 7 days. In general, the Gram positive bacteria remained viable longer than the Gram negative bacteria, perhaps due to the thinner protective peptidoglycan layers of Gram negative bacteria. Additionally, all bacteria remained viable on mobile phones for at least 6 h, which is long enough to be transmitted to a clinical setting and out to the community. It is therefore obvious that mobile phone cleaning/disinfection should occur in combination with proper hand hygiene [55].

In contrast, in terms of the ATP measurement results (refer to Table 10), participants who equally preferred the use of liquid soap and soap bar, followed by those who did not know which of these was more effective, demonstrated higher RLU values compared to the rest of the participants. This finding is an indication that liquid soap could potentially be more efficient in removing organic matter from the hands compared to soap bars, but further research is required for safer conclusions. Similarly, hand washing reluctance ("I do not feel like doing it") as a declared reason for neglecting hand washing also led to higher RLU values on students' mobile phones as expected. Interestingly, a declared hand washing frequency between 6–30 times led to lower RLU values compared to the extremes. Hence, it is safe to say that a moderate daily hand washing frequency is ideally required in order to maintain a low microbial load level on our hands while keeping a healthy and balanced microbiome. However, the fact that most mobile phone samples exceeded the "fail" threshold of 300 RLU remains a cause for concern. Indicatively, in a previous study where the microbial load on mobile devices of healthcare students and nursing staff was assessed, the defined cut-off value was exceeded 10 times by the RLU values of nurses' mobile phone samples, and up to 20 times by the RLU values of the students' mobile phone samples [9].

It is worth mentioning that no correlation between the $cfu/cm^2$ and RLU values was observed in this study. This probably happened because the participants' mobile phones were colonized with plenty of other bacterial and fungal (e.g., *Candida*) species, as well as with ATP-containing organic matter, apart from those isolated by the selective culture

media used in this study, which affected the RLU values. Correlations between $cfu/cm^2$ and RLU have also been previously investigated, leading to mixed results as well. While some reports indicated that high $cfu/cm^2$ values are usually correlated with high RLU values [8], other studies found no correlation between $cfu/cm^2$ and RLU values [13]. Likewise in the present study (Refer to Table 11), while high RLU values were correlated with high $cfu/cm^2$ of *Staphylococcus aureus*, this did not apply in the cases of *Staphylococcus saprophyticus* and *Escherichia coli*.

Participants' field of study had greatly influenced the microbial load on their mobile phones. Undergraduate students of health sciences demonstrated lower RLU values on their mobile phone samples compared to those of students of other sciences (refer to Table 12). In contrast, a higher microbial load of *Escherichia coli* was observed on the mobile phone surfaces of students of health sciences, obviously due to their job's nature, compared to those of students of other sciences. Previous studies also indicated a statistically significant association between the participants' faculty and their mobile phones' microbial load [43], where the isolation of *Escherichia coli* was higher from the touchscreen mobile phones of healthcare workers ($p < 0.05$) [56], due to their job's nature as well. In this study, faculty was a determinant for other declared personal protective behaviors as well, where most students of health sciences considered staying at home and the use of face masks as equally effective, while most students of other sciences did not know which of these was more effective. However, most undergraduate students of other sciences considered liquid soap and soap bars as equally effective, while most students of health sciences preferred the use of liquid soap.

Soap preference was also determined by the participants' year of study (refer to Table 13). The majority of first-year, second-year, and third-year undergraduate students preferred the use of liquid soap, while an equal share of second-year and final-year students considered the liquid soap and soap bar as equally effective. Furthermore, there was an equal share of protective back covers applied to the majority of the mobile phones of students from all years of study. In addition, a higher microbial load of *Escherichia coli* was observed on the mobile phone surfaces of first-year and third-year students compared to those of the second-year and final-year students. This finding indicates that healthcare students are probably becoming more aware of the health hazards and prevention methods at work as they proceed through their curricula. Additionally, it is no coincidence that participants who preferred the use of liquid soap, either based on their faculty or their year of study, demonstrated a higher microbial load of *Escherichia coli* $cfu/cm^2$. However, the mobile phones of those who preferred the use of soap bars demonstrated higher $cfu/cm^2$ of *Staphylococcus aureus*. These contradictory results concerning the efficacy of each soap type against certain bacteria require further investigation by future studies.

Moreover, similar trends were observed between the majority of male and female participants concerning their preference for paper towels, their hand washing duration, as well as the abstaining from touching their face, as a declared personal protective behavior (refer to Table 14). However, male participants provided higher response rates of forgetfulness compared to female participants as a reason for neglecting hand washing, probably due to temperament factors and characteristics of the participants' personality. In contrast, the mobile phone samples of female participants, which tested positive for *Staphylococcus aureus*, were three times more than those of male participants. Finally, it is worth noting that apart from an indication of higher *Staphylococcus aureus* contamination of females' mobile phones, which requires further investigation by future studies, no further statistically significant differences were observed between the microbial load of the participants' mobile phones and their sex. Previous studies also indicated no differences in the $cfu/cm^2$ and RLU values on mobile phones of male and female undergraduate students [8,36].

Generally, the identification of differences in knowledge, attitudes, and behaviors of young adults aged between 18–29 years old on hand hygiene and mobile phone hygiene could be useful in public health policy making and health education planning, since they are considered a heterogeneous social group [43]. To the best of the authors' knowledge,

this is the first study that investigates the knowledge, attitudes, and behaviors of Greek undergraduate students on hand hygiene and mobile phone hygiene, combined with an assessment of the microbial load on their mobile phone devices, based on their sex, faculty, and year of study. According to this study's results, emphasis should be placed on the education of undergraduate students, especially those in their junior years of non-healthcare and healthcare curricula, on hand hygiene and mobile phone hygiene topics. As proposed by other studies [57,58], the inclusion of such courses in their main curriculum enhances the knowledge, attitudes, and behaviors of future healthcare and non-healthcare professionals.

Finally, results should be interpreted with some caution, since the limitations of the present study include the following: the self-reporting of information regarding the participants' hand hygiene and mobile phone hygiene habits [59]; the cross-sectional study design, which does not allow for causality conclusions [60,61]; the convenience sampling method used, which affects the generalizability of these results [62]; the rather small study sample given the number of variables that were analyzed; as well as the exclusive study of specific bacterial species and not any other additional bacterial or fungal species (e.g., *Candida*). Nevertheless, this study provides a clear view and additional information on the students' knowledge, attitudes, and behaviors on hand hygiene and mobile phone hygiene associated with the microbial contamination of their mobile phone devices. Longitudinal data are necessary to further unravel the hand hygiene and mobile phone hygiene habits of undergraduate students, as well as their mobile phones' degree of contamination. Future research should focus on investigating the transmissibility patterns of pathogenic bacteria due to mobile phone use among the social group of undergraduate students, as well as their viability on students' mobile phone surfaces.

## 5. Conclusions

Hand washing is considered by undergraduate students as the most effective personal protective behavior. Despite the majority's appropriate hand washing duration and frequency, there was a considerable number of students who washed their hands inadequately. One major aspect that could potentially improve students' hand hygiene performance is the annihilation of spatial restrictions and forgetfulness. Additionally, given the high rate of bacterial exchange between hands and mobile phones, it is evident that in terms of mobile phone hygiene, students are required to clean/disinfect their phones more frequently. This is because similarly to previous studies, this research proved the abundance of the potentially pathogenic bacterial species of *Staphylococcus saprophyticus*, *Staphylococcus aureus*, and *Escherichia coli* on students' mobile phones. The prevalence of these bacteria is highly associated with certain personal protective behaviors and hygiene preferences. Moreover, faculty was found to be the greatest parameter that determines the microbial load on students' mobile phone devices, followed by their year of study, and their sex. In summary, the knowledge, attitudes, and behaviors of undergraduate students on hand hygiene and mobile phone hygiene are highly intertwined with the microbial load on their mobile phone devices. Therefore, health policy makers, as well as education planners, need to consider adding the concepts of hand hygiene and mobile phone hygiene at an early stage in all undergraduate curricula of health and other sciences.

**Author Contributions:** Conceptualization, P.G. and K.M.; Data curation, D.D. and V.C.; Formal analysis, D.D. and V.C.; Investigation, D.D.; Methodology, P.G.; Project administration, P.G. and K.M.; Resources, P.G.; Software, D.D. and V.C.; Supervision, P.G.; Validation, P.G., V.C. and K.M.; Visualization, D.D.; Writing—original draft, D.D.; Writing—review and editing, D.D. and P.G. All authors have read and agreed to the published version of the manuscript.

**Funding:** This research received no external funding. The APC was funded by the Special Account for Research Grants—University of West Attica and the M.Sc. Program in Public Health—Department of Public Health Policy, University of West Attica.

**Institutional Review Board Statement:** The study was conducted in accordance with the Declaration of Helsinki, and approved by the Research Ethics Committee of the University of West Attica (protocol code 99754/9 November 2021).

**Informed Consent Statement:** Informed consent was obtained from all subjects involved in the study.

**Data Availability Statement:** The research data are available on request by contacting the corresponding author. The data are not publicly available due to privacy restrictions.

**Acknowledgments:** We would like to sincerely thank the Rectorate of the International Hellenic University of Greece for granting permission to conduct our research at the premises of the Alexandreia campus, located in the area of Sindos–Thessaloniki Greece, as well as all of the students who participated in the study. We would also like to sincerely thank the administration of the Occupational Health and Safety Laboratory—Department of Public and Community Health—University of West Attica Greece, especially Konstantinos Ntelezos, for kindly providing us the luminometer device. Furthermore, we would like to express our gratitude and appreciation towards the administration of the microbiological laboratory of Stergiani Giantsiou, for the granted access to the laboratory's equipment and facilities, in order to conduct the microbiological analysis of our samples. Moreover, we would like to sincerely thank the research teams of Głabska et al., Guzek et al. and Cicciarella Modica et al., for allowing us to use part of their research questionnaires' questions for the composition of our study's questionnaire. Last but not least, we would like to thank Yiannis Pittas, for editing the English of this manuscript.

**Conflicts of Interest:** The authors declare no conflict of interest.

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
