# Peer review of "Hygiene Knowledge, Attitudes, and Behaviors of Greek Undergraduate Students on Mobile Phones and Corresponding Devices’ Microbial Assessment"

_2673-947X, doi:10.3390/hygiene3040026_

Round 1

Reviewer 1 Report

On account of the manuscript HYGIENE-2561996, entitled “Knowledge, Attitudes, and Behaviors of Greek Undergraduate Students on Hand and Mobile Phone Hygiene and Assessment of the Microbial Load on Their Mobile Phone Devices” by Dimitrios Delitzakis et al., the authors evaluated the knowledge, attitudes, and behaviors of Greek undergraduate students on hand and mobile phone hygiene based on 100 Greek male and female undergraduate students of health and other sciences. The topic is important to better understanding of the hand and mobile phone hygiene in undergraduate students in Greek, and to conduct human health management as well. After careful consideration, I feel that this manuscript is to be published after improvement of some major shortcomings. Details of my comments are as follows:

1) The manuscript is well written and easy to follow, and the authors got interesting results. Several revisions are, however, required before publication. The first one is in the novelty of the research. Although the authors mentioned the aim of this study, new aspect or view point of this research was not clearly stated in the manuscript. Introduction is not well structured. The authors don't necessarily mention general issues in detail, but are better to show information in a summarized way with focusing on the main issues related to the originality of this study. The authors are strongly encouraged to mention the new viewpoints and/or novel aspects which surpass the previous researches in the manuscript.

2) The present Abstract was not informative. Abstract should include purpose of the research, principal results and major conclusions in a summarized way. In addition, due to separation of the Abstract from the major article, it must be a key to lead readers to evoke a spirit of challenge to contact with the contents of the report. Therefore, the authors are strongly encouraged to improve the Abstract for enhancement of the novelty and better understanding of the results.

2) Another aspect is in the experimental methodologies (validations). Although the authors analyzed Staphylococcus aureus, Staphylococcus saprophyticus, and Escherichia coli in swab samples, the validations of quality assurance (QA) and quality control (QC) for determination of these microbials was not clearly described in the present manuscript. The authors are encouraged to show these results in more detail in the manuscript for enhancement of the accuracy and reliability of the results.

Reviewer 2 Report

Interesting study with important results, especially given current postcovid circumstances and the importance of students as a volatile population group in regards to infection transmission dynamics. 

The rather small study sample, that precludes safe conclusions particularly given the high number of variables that were analyzed, should be also acknowledged in the limitations.

The conclusions section is merely a reproduction of parts of the discussion. I'd suggest authors modify this section in order to outline the implications of their study findings.

English quality is adequate. Some words and phrases are repeated, particularly in the Results section, which isn't actually necessary to do so.

Reviewer 3 Report

The manuscript describes Greek undergraduate students' knowledge, attitudes, and behaviors on hand and mobile hygiene. The authors reported some bacterial contamination on several mobile devices. The study has no novelty and does not add anything to the existing literature. Most of the findings are already established well. The study has the following other major flaws which should be addressed.

1.     The manuscript needs substantial corrections in the English language. Several bacterial and fungal names should be italicized.

2.     One of the major flaws is this manuscript mimics another manuscript entitled “How Did the COVID-19 Pandemic Change the Hand and Mobile Phone Hygiene Behaviors of Greek Undergraduate Students?” https://www.mdpi.com/2673-8112/3/2/20, which has been published by the same authors. It seems that both studies were conducted on the same population (100 undergraduates), same gender distribution (51 males and 49 females), same ethical approval number, and the same duration of study. Both manuscripts target the “Hand and Mobile Phone Hygiene Behaviors of Greek Undergraduate Students”. The only difference is current study considered bacterial isolates, while the published paper was on COVID-19. The methodology and results presentation is also very similar.

3.     2.4.1. Microbiological Sampling and Analysis shows that the authors used Oxoid™ Brilliance Staph 24 Agar for Staphylococci and Oxoid™ Violet Red Bile Lactose Agar, which is a lactose-containing selective medium for the detection and enumeration of Escherichia coli. How did the author know that they would only find these contaminants? Why did not use the culture media/ techniques to isolate other enteric and non-enteric bacterial species which have been reported well in the literature? The study reported less frequently isolated bacteria (Staphylococcus saprophyticus) but what about the most common skin flora often present on the hands and mobile phones?

4.     2.4.2. On-site Microbial Load Assessment lacks clarity. How did the authors assess the bacterial load?

5.     The percentages in Figure 1 are not clear.

6.     The conclusion is unnecessarily prolonged and should focus on the study’s findings.

7.     The manuscript includes most of the older citations, and no data from recent studies have been included.

Needs substantial improvement. 

Reviewer 4 Report

The authors assessed the undergraduate students’ knowledge, attitudes, and behaviors on hand and mobile phone hygiene, along with the microbial load on their mobile phones. The authors narrowed their study to Greek undergraduate students. It is a very promising study, however, the following can help improve the quality:
a) Title should be modified to "Hygiene Knowledge, Attitudes, and Behaviors of Greek Undergraduate Students on Mobile Phones and Corresponding Devices' Microbial Assessment"
b) The introduction needs more information. Please restructure the introduction to flow as follows:
- Microbial entities on human skin, narrowed to hands, and how it spreads onto various contact spaces
- Mobile phones as a career of microbes, the concept of cleanliness of various surfaces, and the challenges therein
- Undergraduate students around the globe, how they handle mobile phones, identify with previous studies performed concerning mobile phones, microbial entities, etc., in Europe. Create a table with three columns, namely a) references, b) objective of the study; and c) key highlights

c) Materials and Methods appears ok. Given the nature of this work, please start the materials and methods with a new subsection captioned "Schematic overview of the study" with a flow diagram compulsorily and text comprise 3 sentences, first should introduce the flow diagram (snapshot of the entire materials and methods), second should link this schematic diagram to the objective of the study, third should indicate whether ethics was required, and if yes, that it was performed in accordance to the required international standard.

c)The results and discussion are ok...Please try to deepen the discourse when discussing results with relevant literature. In this case, kindly apply your discretion. Try to also sharpen the various discussions. Please, in the discussion, kindly make sure that all the places various data from Tables 1-13 (now Tables 2-14), as well as Figures 1-2 (now Figures 2-3) are mentioned in the discussion as " (Refer to Table x) OR (Refer to Figure x)". This will help guide readers to follow adequately all the various reported data, and their discussion.

Please authors, kindly attend to all of the above diligently, because the reviewer will digligently examine the revised version.

Round 2

Reviewer 1 Report

On account of the manuscript HYGIENE-2561996R1, entitled “Hygiene Knowledge, Attitudes, and Behaviors of Greek Undergraduate Students on Mobile Phones and Corresponding Devices’ Microbial Assessment” by Dimitrios Delitzakis et al., the author revised the manuscript appropriately according to the Reviewers comments. After careful consideration, I made a decision that the manuscript is acceptable for publication in its present form.

Reviewer 3 Report

The authors have addressed the comments.